# Speciation Analysis Highlights the Interactions of Auranofin with the Cytoskeleton Proteins of Lung Cancer Cells

**DOI:** 10.3390/ph15101285

**Published:** 2022-10-19

**Authors:** Monika Kupiec, Agnieszka Tomaszewska, Wioletta Jakubczak, Maja Haczyk-Więcek, Katarzyna Pawlak

**Affiliations:** 1Chair of Analytical Chemistry, Faculty of Chemistry, Warsaw University of Technology, Noakowskiego 3, 00-664 Warsaw, Poland; 2Chair of Medical Biotechnology, Faculty of Chemistry, Warsaw University of Technology, Noakowskiego 3, 00-664 Warsaw, Poland

**Keywords:** auranofin, gold speciation, ICP-MS, ESI-MS, lung cancer, tryptic peptide map, TrxR, apoptosis

## Abstract

Two types of lung cells (epithelial cancer lung cells, A-549 and lung fibroblasts MRC-5) were exposed to the clinically established gold drug auranofin at concentrations close to the half-maximal inhibitory drug concentrations (IC_50_). Collected cells were subjected to speciation analysis using inductively coupled plasma mass spectrometry (ICP-MS). Auranofin showed better affinity toward proteins than DNA, RNA, and hydrophilic small molecular weight compounds. It can bind to proteins that vary in size (~20 kDa, ~75 kDa, and ≥200 kDa) and pI. However, the possibility of dimerization and protein–protein complex formation should also be taken into account. µRPLC/CZE-ESI-MS/MS studies on trypsinized proteins allowed the indication of 76 peptides for which signal intensity was influenced by auranofin presence in cells. Based on it, identity was proposed for 20 proteins. Except for thioredoxin reductase (TrxR), which is directly targeted by gold complex, the proteins were found to be transformed. Five indicated proteins: myosin, plectin, talin, two annexins, and kinase M3K5, are responsible for cell–cell, cell–protein interactions, and cell motility. A wound healing test confirmed their regulation by auranofin as cell migration decreased by 40% while the cell cycle was not interrupted.

## 1. Introduction

Various gold(I) and gold(III) complexes belong to a group of rapidly emerging novel therapeutic and diagnostic metal complexes with improved tumor tissue specificity and remarkable antiproliferative potency [1]. They are usually responsible for drastic cancer growth inhibition. One of them, auranofin (AUF, 2,3,4,6-tetra-O-acetyl-1-thio-β-D-glucopyranosato-S) (triethylphosphine) gold(I)) was developed more than 30 years ago as an oral medication for rheumatoid arthritis [2]. Although its use in clinical practice has decreased, studies on auranofin have continued and show promise for treating several diseases, including cancer and bacterial and parasitic infections [2,3,4]. Auranofin is a prodrug metabolized to its pharmacologically active derivatives after administration. In addition to that during the last two decades, repurposing studies revealed that AUF is a promising candidate drug for treating several microbial and parasitic diseases and some forms of cancer (leukemia, non-small cell lung cancer, and ovarian cancer) [5,6]. Still, the mechanism of action for auranofin is not fully understood.

Auranofin consists of two parts: a water-soluble aurothioglucose (GSAu) entity with a sulfur donor group and a hydrophobic phosphine ligand (tEP) which can also form triethylphosphine)gold(I) cation (AutEP) (Figure 1). The AutEP forms adducts with proteins through the coordination of cysteine or selenocysteine residues, especially those in proximity to histidine and with Zn^2+^ binding proteins [7,8,9]. The process can be classified as ligand exchange as the phosphine oxide was reported to be a specific by-product of auranofin’s interactions with human albumin and glutathione [10]. The affinity of AutEP to cysteine, selenocysteine-rich proteins and other purified proteins was confirmed using mass spectrometry. Studies included albumin (Alb) [11], hemoglobin (HlB) [6], selenoprotein P [12] or mammalian thioredoxin reductases (TrxRs) [13,14].

A similar mechanism of action can be expected for gold(I) thioglucose moiety of auranofin (GSAu). Progressively deacetylated thioglucose can be released from the complex by hydrolysis or ligand exchange during adduct formation with protein (Figure 1) [15]. Thioglucose species with a variable number of acetyl groups can form disulfide bridges with proteins.

Like many other cytotoxic drugs, auranofin promotes the activation of the intrinsic apoptotic pathway. As a metallodrug, it was expected to target nucleic acids primarily [16]. However, evidence is growing that auranofin leads to the induction of mitochondria-dependent apoptosis. The intrinsic apoptotic pathway can be triggered by irreparable genetic damage, hypoxia, extremely high concentrations of cytosolic Ca^2+^, and severe oxidative stress [16,17]. Both moieties of auranofin can inhibit thioredoxin reductase (TrxR) and glutathione peroxidase (GPx) [18,19]. As both proteins contain thiol and selenol groups, auranofin’s presence also significantly depletes the reduced form of glutathione [20]. All three biocompounds are responsible for ROS regulation, and their amounts tend to be significantly higher in cancer cells, especially in non-small cell lung carcinoma [21].

Recently, it was shown that auranofin also disrupts intracellular homeostasis of calcium [22], magnesium, and other transition metal ions (Cu, Fe, Zn) [23] involved in reactive oxygen species (ROS) synthesis via the Fenton reaction [24]. It should also be noted that the increased content of metal ions concerned not only the cytosol but also the total cellular content (enhanced accumulation of metal ions by cells). In the next step, pro-apoptotic Bax and Bac proteins from the Bcl-2 family are activated, and anti-apoptotic Bcl-2, Bcl-xL, and Mcl-1 proteins are neutralized [17,25]. Consequently, cells are blocked in the growth phase (G1), unable to proliferate [26,27,28]. Finally, the endoplasmic reticulum responds by inducing changes in mitochondrial membrane permeability, releasing cytochrome C into the cytosol, which activates caspases [29,30]. Consequently, auranofin is perceived as a metallodrug that triggers apoptosis through elevated ROS content.

However, other proteins against which auranofin shows affinity are still being detected and may alter their behavior. For example, AUF was found to activate ribonuclease A (RNase A) and deoxyribonuclease I (DNase I) [7], mediating internucleosomal degradation. Other reports confirmed the inhibition of different enzymatic proteins involved in the apoptosis process, such as IκB kinase [31], protein kinase C iota type [32], hexokinase responsible for glycolytic flux [33], proteasome-associated deubiquitinases (DUBs) [34]. It has been proved that NF-kappa-B protein and CHORDC1 (cysteine and histidine-rich protein) were oxidized next to TrxR in human colorectal carcinoma [35]. Moreover, the affinity of auranofin to many other proteins is still postulated but not proven. Therefore, the mechanism of action for auranofin toward cancer cells is still poorly understood and requires further studies.

The study aimed to investigate the affinity of auranofin toward proteins present in human non-small cell lung cancer (NSCLC, line A-549, epithelial cells [36]) and fetal lung tissue cells (line MRC-5, fibroblasts [37]) to investigate whether these interactions are specific to the cellular proteome. The study was carried out by combining speciation analysis of gold with proteomic strategy. The first utilizes atom-specific mass spectrometry with inductively coupled plasma (ICP-MS), and the second molecule-specific electrospray mass spectrometry (ESI-MS/MS) coupled with different separation techniques. 

ICP-MS (alone or in combination with a separation technique) is used to study the distribution of metal ions [23], design purification and enrichment processes for compounds containing metal ions [12], and study their transformations [38,39]. In the next step, ESI-MS/MS is applied to determine the structure of previously isolated metal compounds or the bioligands themselves (depending on the stability of the complexes/adducts) [12,39]. Both steps fall within the scope of metal speciation analysis applied in metallomics. This type of research can be divided into targeted studies on the interaction of ions/metal compounds with previously isolated and purified bioligands (often standard substances or mixtures mimicking the composition of the serum or cytosol) [38,39] and partially targeted, aimed at detecting new metal compounds formed in a biological object (bioligand-metabolite adducts) or just changes in the composition of the metabolome or proteome [35]. These are, to our knowledge, the first studies of auranofin metabolism in lung cells, not in the isolated cytosol.

## 2. Results and Discussion

Quantitative and speciation analysis was conducted after a qualitative assessment of changes in cell morphology under microscopic observation and estimation of cells’ viability [23]. In addition, the cell density had to reach a minimum of 2 × 10^4^ and 3 × 10^4^ cells (RSD 20%) in 1 mL for MRC-5 and A-549 cells, respectively.

Auranofin, as a prodrug, can be hydrolyzed or oxidized in the blood or cytosol environment before adduct formation. Therefore, its potential metabolites will be designated as auranofin’ (AUF’).

### 2.1. The Total Amount of Gold and Its Fractionation

We have previously reported that cancer and normal cells accumulate up to 10% of auranofin. Special care is needed when collecting auranofin-treated cancer cells, as an adhesion for up to 15% of cells was reduced. They were gently harvested and centrifuged, and the gold content taken up by the cells from the medium was then determined. Up to 38.1% of the gold was present in cancer cells isolated from the medium without trypsinization. The remaining 61.9% of the gold was determined in adherent cells. Interestingly, in the case of normal cells, gold was mainly found in adherent ones. The higher number of non-adherent cancer cells is a characteristic response to auranofin exposure. 

We performed liquid-liquid extraction according to the procedure published by Chomczynski [40] to separate proteins and hydrophilic compounds from hydrophobic ones. When a phenol-chloroform-water mixture at pH 7 was added to the lysate after centrifugation of the sample, three layers were obtained, as expected:

aqueous—containing hydrophilic, usually low-molecular-weight compounds, plus RNA and DNA

organic—containing lipids, and other hydrophobic compounds

interphase—containing precipitated proteins.

Most of the gold was detected in the organic fraction (up to 65%, Figure 2). Auranofin metabolites may be attached to lipids, but it cannot be ruled out that this fraction also contains poorly water-soluble auranofin metabolites. From 11.1 to 18.9% (SD 0.2%) of gold was detected in the water phase containing water-soluble metabolites and DNA/RNA. The gold content of this fraction was twice as low as that of platinum when the same experiment was performed with cisplatin (Appendix A), probably due to the high solubility in water of hydrolyzed platinum complex. Up to 24.3% of gold was present in the interphase-containing proteins (a higher amount in MRC-5 cells). Auranofin shows a better affinity to proteins than DNA/RNA in contrast to cisplatin in cancer cells. In normal cells, platinum compounds were present in non-adherent cells and were evenly distributed (35%, 37% and 27% of platinum was determined in the fraction containing hydrophobic, protein and hydrophilic compounds, respectively). This is in agreement with the results of other research teams showing that cisplatin, in addition to its affinity for DNA [41], forms stable adducts with cysteine- and methionine-containing proteins [42,43], shows an affinity for copper transport proteins and affects the structure of the hydrophobic plasma membrane (PM) [44]. Both metal complexes cisplatin and auranofin show affinity for a similar group of compounds in A-549 cells. In the case of fibroblasts (MRC-5), the gold content was detected proportionally to the number of adherent and non-adherent cells (92% of the total cell population consists of adherent cells that contain 98% of total accumulated gold). Unlike platinum content concentrated in non-adherent cells (50% of the total cell population consists of non-adherent cells that contain 99% of total accumulated platinum). This indicates the homogeneous accumulation of AUF’ by the fibroblast cells and the drug-induced uptake of cisplatin. Cisplatin was proven to interact with proteins and lipids, influencing membrane composition, fluidity, and permeability [44,45]. Such alteration of PM can enhance metallodrug diffusion and uptake. It depends on the cell type due to differences in PM and ECM composition [45] (Appendix A).

### 2.2. Confirmation of the Presence of Auranofin’-Protein Adducts by SEC-ICP-MS and CE-ICP-MS

The affinity of auranofin to proteins in cell lysates was investigated by size exclusion chromatography (SEC), and capillary zone electrophoresis (CZE) coupled to mass spectrometry with inductively coupled plasma (ICP-MS). SEC allows compounds to be distinguished by molecular weight differences, while CZE separates compounds by mass-to-charge ratio differences. 

In the SEC chromatograms, it can be seen that AUF’ forms adducts with proteins in cancer cell lysates, which is observed as three peaks: (peak 1) of molecular weight ≥ 200 kDa, (peak 2) around 75 kDa, and (peak 3) 20 kDa (Figure 3a). Only two peaks (2 and 3) were observed in the chromatograms obtained for lysates of MRC-5 cells exposed to auranofin (Figure 3a). The molecular weight of AUF’-adducts was determined based on SEC column calibration using a commercially available protein mixture (Protein Standard Mix 15–600 kDa and UV detection) and additionally verified by comparing the retention times obtained for AUF’-adducts with proteins such as albumin (69 kDa), holotransferrin (85 kDa), cytochrome c (17 kDa), metallothionein (8 kDa) and glutathione (<1 kDa).

For capillary electrophoresis, the migration time increases with the molecular weight of compounds when they migrate as negatively charged compounds (behind the negative methanol peak at 5.6 min—the EOF marker) and the opposite when positive (Figure 2b).

Auranofin, a weakly water-soluble compound, migrated close to the EOF marker. The samples were ultrafiltered (10 kDa cut-off filtrates) shortly before analysis to ensure that the observed peaks corresponded to AUF’-adducts with non-small molecular weight compounds (especially glutathione) [46]. The electropherograms did not differ, so small molecular weight compounds (e.g., free AUF metabolites) were excluded from further discussion. Peak 1 obtained for cancer cells can correspond to an adduct of AUF’ with cytochrome c or other protein with pI >> 7.0. Peaks 3 and 4 can correspond to an adduct formed by protein with pI < 7.0. The resolution of the CZE method appeared to be similar to that obtained by SEC as, again, three peaks were observed for the cancer cell lysate and one for the MRC-5 cells lysate (Figure 3b). 

Lysates from control groups spiked with auranofin were analyzed by both separation techniques. This procedure verifies whether bioligands form the adducts in the cells or whether their quantity depends on the presence of auranofin. Changes in the profile of chromatograms were negligible for MRC-5 cells. Only one additional peak was obtained in the CZE electropherogram. In the case of cancer A-549 cells, the chromatograms and electropherograms obtained for auranofin-treated cells differed significantly from those obtained for control group lysates spiked with auranofin (additional SEC peaks 1 and 2 and CZE peaks 1 and 4 (Figure 3)). Such a change can be observed as a result of upregulation/modification of auranofin’ binding proteins or changes in the cell’s environment enhancing metal complex binding. However, we expected an adduct with a molecular weight of about 70 kDa to be observed as the concentration of TrxR is higher in A-549 cells due to oxygen stress.

SEC-ICP-MS chromatograms for lysates of cells exposed to cisplatin and control sample spiked with cisplatin consisted of one peak only corresponding to protein of molecular weight 10 kDa (Appendix A). This indicates much less significant changes in the composition of the proteome compared to those induced by auranofin.

As both separation techniques confirmed the presence of AUF’-protein adducts and significant changes in proteome composition, tryptic digestion can be performed to characterize it in detail.

### 2.3. Au-Tagged Tryptic Map Obtained by CZE/µRPLC-ICP-MS

The CZE-ICP-MS tryptic peptide maps (Figure 4a) differ significantly from those obtained for proteins in lysates (Figure 3b). However, these changes do not confirm that all the proteins have been enzymatically digested. Still, the electropherograms obtained with ICP-MS detection can indicate the migration time of gold-binding compounds.

Reversed-phase chromatography trypsin maps (Figure 4b) were obtained using gradient elution (see Section 3.8 for conditions). Four peaks were obtained for A-549 cancer cells and two for MRC-5 normal cells (Figure 4b). The recovery of gold was 89–92% for A-549 and MRC-5 digests, respectively. It was established as the relative value against the total peak area obtained by analysis of the same sample using µHPLC-ICP-MS in the FIA mode.

Since trypsin is a protein containing ten cysteine residues, its enzymatic activity may be reduced due to interaction with metal ions (via ligand exchange). Therefore, it is necessary to verify the efficiency of enzymatic digestion. Extracts were also analyzed with µRPLC-ICP-MS before enzymatic digestion to compare with tryptic maps. The first peak was not observed until 50 min, while non-digested lysates were analyzed even when 95% acetonitrile was used during the last 20 min. Since the chromatographic tryptic map was completed within 20 min, all the proteins present in the sample were enzymatically digested. It is, therefore, possible to determine the peptides’ identity using LC-ESI-MS/MS.

### 2.4. Identification of Auranofin Adducts with Peptides by CZE/µRPLC-ESI-MS/MS

All samples, including the blank (*n* = 3), were analyzed by µRPLC-ESI-MS and CZE-ESI-MS in the scanning mode (Figure 5).

In the next step, mass spectra were sequentially summarized for the 60 s range (20 mass spectra for each CE and 30 for each RPLC analysis). This procedure was necessary to avoid signal loss due to noise subtraction from the entire chromatographic range. The subtracted mass spectra were thoroughly searched for signals:-(a) specific for MRC-5 cells (mass spectra for MRC-5 cells subtracted with mass spectra obtained for A-549 cells)-(b) specific for A-549 cells (obtained with sequence opposite to point (a))-(c) lost due to auranofin presence in cells (mass spectra of control sample subtracted with mass spectra obtained for appropriate cell line exposed to auranofin)-(d) arriving due to auranofin presence in cells (obtained with sequence opposite to point (c))

Only signals higher than 1000 cps were selected for extraction of chromatograms (EIC). The signal was excluded from further analysis if a peak was not present in the extracted chromatogram. Otherwise, MS/MS fragmentation was performed using three collision energies (15, 30, 50 eV) to obtain the sequence of amino acids. 

Protein identities were searched using MS/MS data with the MASCOT search engine online using the UniProtKB database (SwissProt 2022_03) limited to Homo Sapiens for peptide mass tolerance ± 0.3 Da. They were proposed if at least two peptides were sequenced. As results were based on low-resolution data, a validation protocol was developed to evaluate qualitative data. It was checked whether the proposed proteins were already found as significantly abundant in non-small lung cancer cells (A-549) using high-throughput mass spectrometry proteome analysis [47,48,49]. Proteins upregulated by auranofin (e.g., Bax, Bim [18]) have been added to the verification list. Additionally, ion chromatograms assigned to each protein were extracted and checked for consistency (Figure 5a–f for selected proteins). Moreover, the correlation of retention times with hydrophobicity (HR) and isoelectronic point (pI) calculated using the ProtPI calculator was verified (r^2^ = 0.71) (www.protpi.ch/Calculator/PeptideTool, accessed on 31 May 2022, Appendix A).

It is noticeable that chromatograms reflect four types of changes in proteome compositions proposed to search m/z signals specific to our experiment. For example, higher and more numerous peaks were observed for PLEC protein in cancer cells (Figure 5c,d) and lower for K1C18 and TLN1 (Figure 5a,b,e,f). Auranofin was responsible for significant signal reduction for 20 proteins (their identities are presented in Table 1) due to their downregulation or chemical interactions interfering with their trypsinization. Changes in extracted chromatograms for identified proteins were similar to those presented in Figure 5a–f. New signals for 20 different peptides were also found by µRPLC/CZE-ICP-MS after exposure of cells to auranofin (Figure 5g–j).

A lower number of peptides detected by capillary electrophoresis (59 out of 76) coupled to ESI-MS compared to µRPLC-ESI-MS were most probably caused by a much smaller volume of sample (100 nL versus 5 µL).

Next, chromatographic peak areas were collected for each parent ion separately with assigned amino acid sequence and proposed protein ID to study its changes in tryptic digests for proteins extracted from A-549 and MRC-5 cells exposed to auranofin. 

### 2.5. Changes in Peptide Composition in the Cell Lysate Due to the Presence of Auranofin

Obtained peak areas were transformed to the relative one calculated as the ratio of a single peak area against the sum of peak areas of one analysis. Next, data was transferred to Clustvis online software for principal component analysis (PCA), cluster analysis, and data visualization.

Incubation of cells with auranofin reduced peak areas for 55 peptides representing 20 proteins, illustrating significant changes in the composition of the lysates. From the PCA plot and heat map (Figure 6a,b), it can be seen that peptide maps of the two cell types differed significantly, but the difference decreased with auranofin-treated cells (Figure 6b). This is because, using our method, we detected more peptides for which signals were lost due to auranofin’s activity than new ones gained. Most peptides detected in samples after treatment of cells with auranofin contained mainly oxidized methionine reflecting the “folding stability” of proteins as Walker postulated [50]. It can explain why three downregulated proteins do not contain cysteine residues (Table 1). 

It should be stressed that various AUF’ species can be bound to thiol or selenol groups. That creates the possibility that different species can be eluted in one µRPLC-ICP-MS peak. Therefore, they are much more challenging to detect, especially with the µRPLC-ESI-MS method. The height of the chromatographic peak for monoisotopic Au obtained by sensitive ICP-MS was only 1 × 10^4^ cps. Therefore, detection of signals derived from AUF’-peptide adducts by µRPLC-ESI-MS is unlikely. Due to that, no selenocysteines were detected either. 

Combining results from SEC-ICP-MS and µRPLC-ESI-MS/MS chromatograms, only two proteins (B2L11, 22 kDa Bim protein, and TRXR1, 71 kDa) can be proposed to bind gold species in MRC-5 cells. Cytosolic cytochrome C cannot be excluded, but only one peptide corresponding to the amino acid sequence for this protein was found in the lysate of MRC-5 cells. A more significant number of proteins that can interact with auranofin were detected in A-549 cells, such as annexins (A1, 39 kDa and A4, 36 kDa), spectrins (α, 284 kDa and β, 274 kDa), talin 1 (270 kDa), chaperonin 60 (58 kDa) and above all selenoprotein thioredoxin reductase 1 (TRXR1, 71 kDa). 

The last protein is known to be upregulated in A-549 cells. It should be recalled that no compounds were eluted (SEC-ICP-MS) at the time corresponding to adducts with 70 kDa proteins in the case of the lysate spiked with auranofin, even after incubation for 24 and 48 h. This means that auranofin modulates the ability of proteins to bind complexes and gold ions or requires the presence of other compounds to enhance adduct formation in the unfriendly environment of the cancer cell. It could also mean that there are no active thiol groups in proteins extracted from the control group of A-549 cells. For this reason, the lysate obtained from cells exposed to auranofin was also enriched with auranofin. It was found that the signals from peaks 1 and 2 increased significantly. Therefore, the deficiency of active thiol groups is not due to the sample preparation method.

Despite the increase in ROS level [23], only two identified proteins are directly associated with oxidative stress (thioredoxin reductase 1, TRXR1, and phosphoglycerate kinase 1, PGK1). Other proteins might not interact directly with AUF’, but its presence interfered with their amount or structure. Usually observed as the signal drop for representative peptides (Figure 6d). Since flow cytometry showed no changes in the cell cycle due to auranofin presence (Figure 6e), changes in abundance for this protein must be induced by auranofin. The method was also applied to cells exposed to cisplatin with significant changes in the cell cycle (Appendix A).

Most identified proteins are responsible for cellular cytoskeleton organization, cell junction, adhesion, or motility. These are annexins 1 and 4 (ANXA1/4), keratins (K1C18 and K2C8), prelamin (LMNA), plectin (PLEC), spectrin (SPTB2 and SPTN1), talin (TLN1) and vimentin (VIME). Higher signals for keratin are commonly observed in epithelial A-549 cells. Keratins can form intermediate filaments to strengthen epithelial cells and protect against oxidative stress and cell death [51,52]. Spectrin and vimentin can bind to actin, calmodulin, and cadherin, thereby modulating the cytoskeleton structure and the integrity of the cell membrane. They can also transmit membrane receptor signals to the nucleus [53]. Prelamin present in the membrane of mitochondria can prevent the release of cytochrome C from mitochondria. Its imbalance with the lamin can increase oxidative stress [54].

Myosin (MYH-9), plectin (PLEC), talin (TLN1), and annexins are responsible for cell–cell and cell–protein interactions accompanied by kinase M3K5 make up the system responsible for cytoskeleton reorganization essential for the cell motility and the wound healing process [20,55]. The abundance of peptides representing those proteins significantly decreased due to the exposure of cells to auranofin. Their inactivation was confirmed by the wound healing test, which showed that the movement of epithelial cells was reduced by 40% in the presence of auranofin compared to control samples (Figure 6c). The test was also carried out for cisplatin (migration of cells reduced by 10%, Appendix A). 

Undoubtedly, auranofin induces the mechanism of mitochondrial apoptosis by increasing ROS due to the inactivation of the TrxR protein. However, the influence of auranofin on the behavior of cytoskeleton proteins essential for regulating mitochondrial redox activity and cell movement cannot be ruled out. Cisplatin, which is also responsible for the induction of apoptosis, does not affect cell motility. Most of the peptides, which intensity was influenced by auranofin presence, represented cytoskeleton proteins. The cell cytoskeleton forms a complex system responsible for signal transduction, regulation of cellular transport, mitochondrial activity, morphology, and rearrangement [56,57]. It cannot be ignored, especially since it can explain why cells accumulate metal ions, even in the presence of a metallodrug.

## 3. Material and Methods

### 3.1. Instruments

Size-exclusion chromatography and reversed-phase chromatography were carried out using HPLC 1200 system, and capillary electrophoresis using G1600AX 3D CE System (Agilent Technologies, Waldbronn, Germany). Two types of detection systems were used: (1) molecule-specific electrospray mass spectrometer LC-MS/MS (model 6460 TripleQuad Jet Stream equipped with an autosampler, Agilent Technologies, Santa Clara, CA, USA) and (2) element-specific mass spectrometer with inductively coupled plasma (ICP-MS, model 7500, Agilent Technologies, Tokyo, Japan). Capillary electrophoresis was coupled to ESI-MS using the Agilent interface and to ICP-MS using the MCN-100 interface (CETAC, Omaha, NE, USA). 

### 3.2. Preparation of Growth Media and Cells

As a growth medium, the Minimum Essential Medium Eagle (MEM, Sigma-Aldrich) commercial solution was enriched with 1 mM of penicillin, streptomycin, and 0.25 mM L-glutamine (Sigma-Aldrich). Two MEM-enriched solutions were used: one with 10% Fetal Bovine Serum (FBS-MEM) and one without (MEM, Sigma-Aldrich). Human epithelial lung cancer cell line (A549, CCL-185, passage number: 1) and fibroblast lung cell line (MRC-5, CCL-171, passage number: 30) were purchased from the American Type Culture Collection (ATCC). In FBS-MEM, cells were maintained as a monolayer culture at 37 °C in a humidified 5% CO_2_ atmosphere. Cells were sub-cultured according to the protocols recommended to the cell’s phenotype and described previously in Jakubczak et al. [23]. Cells were detached from the culture flasks using 2 mL of 0.05% trypsin-EDTA solution (trypLE Express, Gibco) for 5 min, suspended in 6 mL of FBS-MEM, and centrifuged for 5 min at 1500 rpm (Universal 32 Tabletop Centrifuge, Hettich, Tuttlingen, Germany). The supernatant was removed, and cell pellets were resuspended in the proper amount of FBS-MEM to obtain the cell density of 1 × 10^5^ cells/mL.

### 3.3. Preparation of Metallodrugs Solutions

Auranofin (Figure 1a) of purity ≥ 98% was purchased from Sigma-Aldrich. A stock solution of auranofin (1.0 × 10^−4^ M) was prepared by dissolving 0.4 mg of auranofin in 100 µL DMSO and 5.9 mL of MQ-water. Next, it was diluted to obtain the solution of 0.6 µM auranofin in MEM, which was protected from light. 

### 3.4. Cells’ Exposure to Metallo-Drugs

Cells were transferred to the flasks with cultivating area of 25 cm^2^ and incubated for 24 h at 37 °C in a humidified atmosphere containing 5% CO_2_ (incubator, HeraCell, ThermoFisher Scientific, Langenselbold, Germany). After incubation, the medium was removed from the flasks, and cells were washed twice with 1 mL of PBS. MEM solution with auranofin was added to the culture flask containing 1 mln cells. The cultures exposed to auranofin and from the control group were kept in the dark for 72 h at 37 °C in a humidified atmosphere containing 5% carbon dioxide. The samples were prepared in four sets for cytotoxicity tests, staining of cells (described previously in Jakubczak et al. [23]), metal determination, and metallo-proteomic LC/CE-MS analysis.

### 3.5. Collection of Cells toward Metal Determination

MEM solution, containing cells dissociated from the culture flask, was gently transferred from the flasks into 15 mL falcons. Remained adherent cells were treated with trypLE Express (Gibco) solution for 5 min. After that, cells were rinsed, suspended in 2 mL of MEM, and added to 15 mL falcons. The obtained cells were centrifuged at 1500 rpm for 5 min. The supernatants were removed, and the cells’ pellets were subjected to lysis. 

### 3.6. Cells’ Lysis, Mineralization, and Fractionation

Cells were lysed with 2 mL of deionized water by means of sonication using a glass probe for 20 min with 30% of maximum power in 4 cycles. The obtained solutions were centrifuged for 20 min at 18,000 rpm at 4 °C twice. The purified supernatant was filtered with a 0.45 µm syringe filter (Sigma-Aldrich, Bellefonte, PA, USA), two first drops were discarded, and only the remaining part of the filtrate was subjected to further steps of sample preparation. Samples for each cell type treated with auranofin and from the control group were subjected to (1) mineralization, (2) liquid–liquid extraction, and (3) fractionation using ultrafiltration before gold determination.

Both types of cells were subjected to microwave-assisted mineralization with a mixture of 5 mL of aqua regia. The digests were diluted to a final volume of 25 mL with MQ water. Further dilutions of the digests or lysates toward ICP-MS analysis were prepared using 2% HCl solution and Rh (10 ng mL^−1^) as an internal standard. 

0.8. mL of phenol-CHCl_3_ (1:1) mixture was added to the cell’s lysate (0.8 mL) [58]. The samples were vortexed for 10 min, centrifuged (18,000 rpm for 10 min), and placed on ice. Three fractions: water, organic, and protein disc, were transferred to separated vials and mineralized with aqua regia.

Another portion of the supernatant solution (0.5 mL) was ultrafiltrated using 10 kDa cut-off filters with 6000 rpm for 20 min at 25 °C. The fraction of high-molecular-weight compounds (HMWC) was recovered by reverse ultrafiltration at 3200 rpm for 15 min. Both high- and low-molecular-weight-compounds (LMWC) fractions were obtained in seven sets. Two sets were transferred to separated vials and mineralized with aqua regia. Others were used for protein digestion.

### 3.7. Digestion of Proteins in High-Molecular-Weight-Fraction Obtained by Ultrafiltration

To 350 or 400 µL of the HMWC fraction obtained from lysate solution (obtained from the cytosol of control and auranofin-treated cells, respectively), a dithiothreitol (DTT, reducing agent) was added to obtain 10 mM final concentration. The mixture was then incubated for 30 min at 37 °C. Next, an appropriate volume of iodoacetamide (IAM for alkylation of cysteine residues) solution was added to obtain the final 50 mM concentration. The mixture was incubated at room temperature for 45 min while protecting the solution from light. After that 10 mM trypsin solution in 10 mM ammonium acetate (pH 7.5) was added to the sample in a ten to twenty-fold excess [59] relative to the total protein concentration determined in the sample using Bradford Assay according to the commercial protocol [60]. The mixture was shaken for 20 h at 37 °C. 

The solution, after enzymatic digestion, was subjected to partial lyophilization, followed by ultracentrifugation of the solution at 10,000 rpm for 30 min at 25 °C. The filtrate containing tryptic peptides (products of the enzymatic digestion of proteins) was analyzed by RPLC/CE-ICP/ESI-MS/MS.

### 3.8. Separation and Detection Conditions

Determination of metal was carried out using ICP-MS. Plasma power, voltage for ion lenses, and torch position were established daily to obtain the highest sensitivity and oxide formation below 0.4% using a standard mixture of 10 ng mL^−1^ Li, Co, Y, Ce, and Tl in a 2% nitric acid solution. Additionally, an isotope of ^181^Ta was monitored in the samples, revealing that the contribution of TaO to the ^197^Au isotope signal used for quantification was not significant. Flow-injection analysis (FIA) was used instead of direct sample introduction due to the small volumes of cells’ lysates. A 50 µL sample was introduced into a stream of mobile phase (5 mM ammonium nitrate) delivered by an HPLC pump to ICP-MS through the Rheodyne valve and PEEK tubing [23]. 

The formation of protein adducts with auranofin was confirmed by capillary zone electrophoresis and by size exclusion chromatography coupled with ICP-MS using a Superdex 200 column (Amersham 30 × 2 cm, 20 µm). A 50 µL sample was injected via a manual injection system, and an aqueous solution of 20 mM Tris-H_3_PO_4_, pH 7.4, was used as the mobile phase. 

Tryptic peptides were separated using a capillary HPLC system coupled with ICP-MS using CETAC-100 micro-concentric nebulizer and conical spray chamber and to ESI-MS using a commercial nebulizer (Agilent Technologies) with a narrower capillary for mobile phase flow up to 50 µL/min. Twin tryptic peptides were obtained using capillary electrophoresis coupled to ICP-MS using a micro-concentric nebulizer with a conical spray chamber and interface cross-connection (CETAC) and commercial nebulizer with an additional connection for makeup liquid. 

Optimal parameters for the separation and detection of peptides and their adducts with gold complex are presented in Table 2.

Total amounts of metals and amounts of metals in lysates were calculated using Chemstation 10.02.B by the internal standard correction method and recalculated against cell density as the normalization factor. Statistical analysis of data was carried out by means of Statistica Software 13.1. Chromatographic and electrophoretic peak areas were obtained using Chemstation 10.02.B and Agilent Mass Hunter Qualitative Analysis B.05.00 software (Agilent Technologies) for signal integration. Collected peak areas were normalized chromatographically against the total peak area obtained for all chromatograms. Relative peak areas were transferred to ClustVis software for data PCA and cluster analysis and its visualization in the form of a heat map [61].

### 3.9. Cell Cycle Analysis and Wound Healing Test

MRC-5 and A-549 cells were incubated with 0.4–0.7 μM auranofin for 24, 48, and 72 h (at a density of 5 × 10^5^ cells/mL). Cells were trypsinized for cell cycle analysis and then washed twice with cold PBS. Cells were centrifuged for 5 min at 1500 rpm. Next, cells were resuspended with 580 μL PBS solution containing 10 μg/mL of DAPI (solution 10, Chemometec), incubated for 5 min, and analyzed by flow cytometry. Cell cycle analysis was performed using Nucleo Counter^®^ NC-3000^TM^ system (Chemometec).

Cells (2 × 10^5^ cells/well) were seeded in 24-well plates to grow in a monolayer for 24 h. Then, a sterile 2–20 μL pipette tip was held vertically to scratch a cross in each well. The detached cells were removed by washing with 500 μL PBS and shaking at 500 rpm for 5 min. 500 μL of fresh medium with auranofin (in the range of 0.4–0.8 µM to achieve a concentration equal to IC50 and in the range of 0.8–1.6 µM to achieve a concentration equal to 2 × IC50) was added to each well and incubated for 72 h. Before the image acquisition, the plate was washed with 500 μL pre-warmed PBS [62] and gently shaken for 30 s. Then, a medium was added again, and pictures were taken. The scratch closure was monitored and imaged in 24 h intervals using an SZX10 microscope (Olympus, Tokyo, Japan). Images were analyzed using ImageJ software with Wound Healing Tool.

## 4. Conclusions

Auranofin is known to induce apoptosis by inactivating the thioredoxin reductase protein (TrxR) responsible for controlling reactive oxygen species (ROS) in the cytosol. Nevertheless, new reports appear indicating that this gold(I) complex may interact with other proteins related to the apoptotic process. Based on the ICP-MS determinations of gold in LLE fractions, it was shown that auranofin has a greater affinity to proteins than DNA, RNA, and hydrophilic low-molecular compounds. All cells absorb auranofin equally, unlike cisplatin, which can enhance its uptake by increasing plasma membrane permeability. However, this effect depends on the cell composition, as it was only observed in fibroblasts (MRC-5). 

Size-exclusion chromatography with inductively coupled plasma mass spectrometry (SEC-ICP-MS) provided chromatograms showing that auranofin can bind more than one protein. They differ in size (~20 kDa, ~75 kDa, and ≥200 kDa) and pI (migrated as positive and negative ions in a background electrolyte buffer pH 6.9). In epithelial cancer lung cells already under severe oxidative stress, their synthesis/activation is induced by auranofin. 

Analysis of the tryptic peptide map showed that auranofin metabolites could bind to peptides. However, their amount was too low to provide appropriate ESI-MS/MS fragmentation data to propose structures for gold-containing compounds. Based on the RPLC-ESI-MS/MS results, we proposed a sequence of amino acids for 76 peptides and identities for 20 proteins. TrxR was also identified among these proteins. However, most of the proteins were associated with the cellular cytoskeleton regulating the activity of mitochondria, cell morphology, reorganization, and influencing cell motility. The last one was confirmed by a wound healing test (the migration rate for cells exposed to auranofin dropped 30–40%) despite not having an interrupted cell cycle. 

In light of these findings, it is very probable that auranofin can affect the behavior of cytoskeleton proteins and, to a lesser extent, the plasma membrane (compared to cisplatin). Obtained results demonstrate the importance of complementary metallomic studies (including changes in the composition of the cellular proteome, metabolome and lipidome) aiming to characterize the activity of metallodrugs.

## Figures and Tables

**Figure 1 pharmaceuticals-15-01285-f001:**
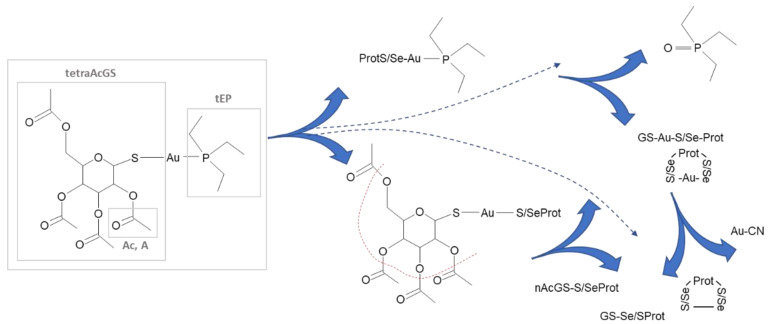
Structure of 2,3,4,6-tetra-O-acetyl-1-thio-β-D-glucopyranosato-S) (triethylphosphine) gold(I) known as auranofin and most commonly reported/postulated products of its transformations.

**Figure 2 pharmaceuticals-15-01285-f002:**
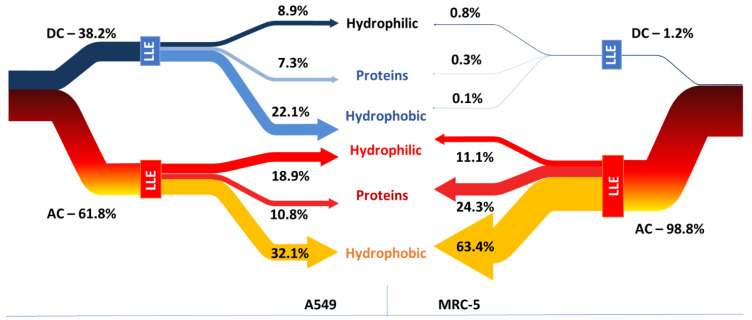
Sankey’s graphs showing the distribution of gold (a characteristic component of auranofin) in three fractions obtained by LLE extraction carried out for adherent (AC) and non-adherent (DC) MRC-5 and A-549 cells.

**Figure 3 pharmaceuticals-15-01285-f003:**
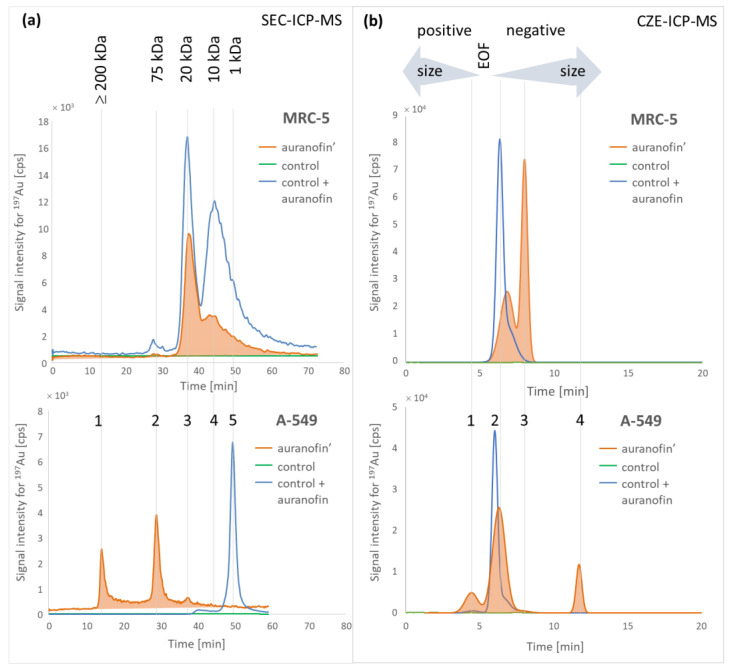
Chromatograms (**a**) and electropherograms (**b**) obtained for lysates of MRC-5 and A-549 cells exposed to 0.6 µM auranofin (auranofin’) for the control group (control) and control group spiked with 0.03 µM of auranofin and incubated for 72 h (control + auranofin).

**Figure 4 pharmaceuticals-15-01285-f004:**
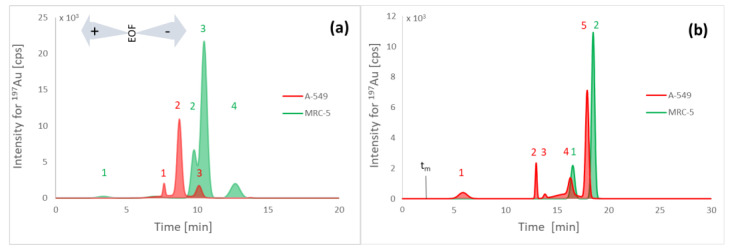
Au-tagged tryptic peptide maps obtained by CZE-ICP-MS (**a**) and µRPLC-ICP-MS (**b**). Electropherograms and chromatograms for the control sample are not shown because only the baseline was observed.

**Figure 5 pharmaceuticals-15-01285-f005:**
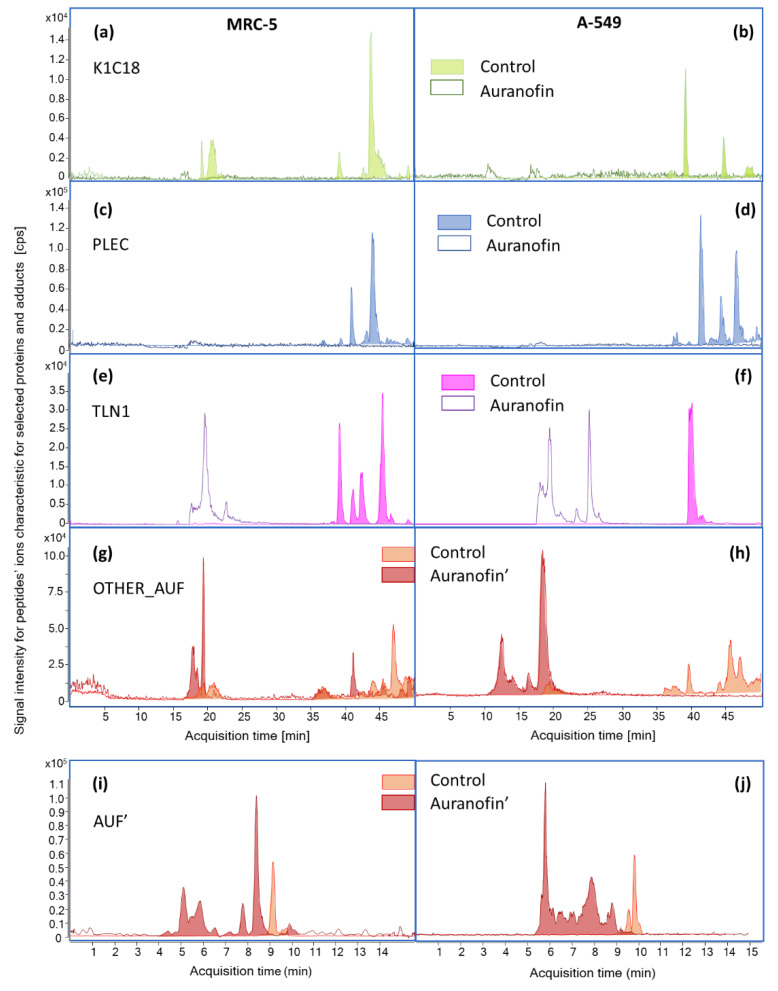
Extracted ion chromatograms (µRPLC-ESI-MS, A-H) for at least three peptides (summed EICs) obtained for selected proteins with proposed identities for MRC-5 (**a**,**c**,**e**) and A-549 (**b**,**d**,**f**) cells in control group and exposed to auranofin. Extracted chromatograms and electropherograms (µRPLC-ESI-MS, (**g**,**h**) and CZE-ESI-MS, (**i**,**j**)) for six signals found in MRC-5 and A-549 cells after exposure to auranofin.

**Figure 6 pharmaceuticals-15-01285-f006:**
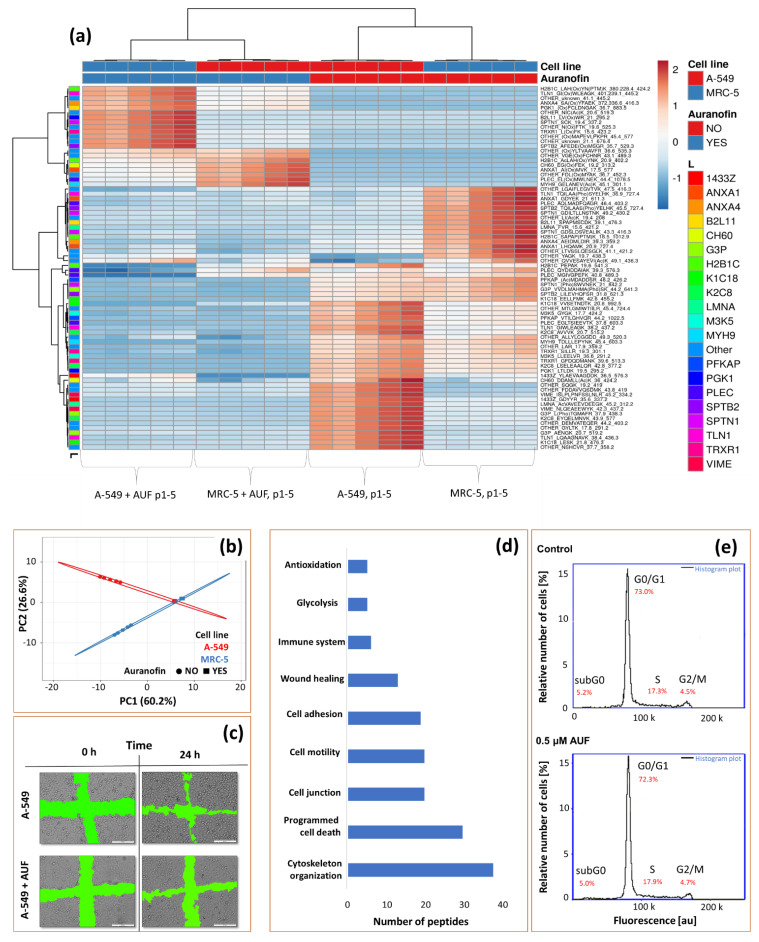
Heatmap (**a**) showing changes in the peptide’s composition registered by µRPLC-ESI-MS related to 20 proteins due to the activity of auranofin. Principal component analysis, PCA chart (**b**) shows the main differences between A-549 and MRC-5 cells before and after auranofin exposure. Images (**c**) present the influence of auranofin on cell motility of A-549 cells (wound healing test). Chart (**d**) presents the number of peptides related to proteins with specified functions. Graphs (**e**) present no influence of auranofin on cell cycle.

**Table 1 pharmaceuticals-15-01285-t001:** The list of proposed identities for proteins based on LC-MS/MS results for tryptic digests for which signal intensity significantly changed for cells exposed to auranofin. The coverage describes the degree to which the amino acid sequence in peptides is defined relative to the protein total sequence. See Appendix A for detailed information.

Protein	Monoisotopic Mass, Da	No Cys	No of RPLC-ESI-MS/MS Peptides	No of CZE-ESI-MS/MS Peptides	COVERAGE [%]
1433Z_HUMAN (P63104)	27,728	3	2	1	6.6
ANXA1_HUMAN (P04083)	38,559	4	3	1	4.0
ANXA4_HUMAN (P09525)	35,729	4	2	2	5.3
B2L11_HUMAN (O43521)	22,157	4	2	1	6.8
CH60_HUMAN (P10809)	57,927	3	2	1	2.5
G3P_HUMAN (P04406)	35,899	3	3	3	7.5
H2B1C_HUMAN (P62807)	13,766	0	4	3	20.5
K1C18_HUMAN (P05783)	47,897	0	3	3	5.0
K2C8_HUMAN (P05787)	53,671	0	3	3	5.2
LMNA_HUMAN (P02545)	73,764	5	3	3	4.0
M3K5_HUMAN (Q99683)	154,440	23	2	2	0.8
MYH9_HUMAN (P35579)	226,261	22	2	2	0.9
PFKAP_HUMAN (Q01813)	85,542	16	2	1	2.2
PGK1_HUMAN (P00558)	44,455	7	2	1	3.3
PLEC_HUMAN (Q15149)	531,466	35	5	3	1.1
SPTB2_HUMAN (Q01082)	274,308	14	3	2	1.4
SPTN1_HUMAN (Q13813)	284,364	14	4	3	1.3
TLN1_HUMAN (Q9Y490)	269,599	38	4	3	1.5
TRXR1_HUMAN (Q16881)	70,862	17	3	3	2.9
VIME_HUMAN (P08670)	53,488	1	2	2	5.4

**Table 2 pharmaceuticals-15-01285-t002:** Optimal parameters for separation and detection of peptides and their adducts with gold complex.

Detection Conditions			
ICP-MS Detection		ESI-MS Detection	
Parameter	HP7500a	Parameter	6460 3Q Jet Stream
Plasma power	1310 W	Nebulization voltage	2500 V
Double charged	0.1%	Nozzle voltage	500 V
Nebulizer gas flow	1.1 L min^−1^	Flow, pressure, and temperature of nebulizing gas	7 L min^−1^, 30 psi, 300 °C
Nebulizer	Cross-flow, CFN	Flow and temperature of the sheath gas	5 L min^−1^, 250 °C
Mobile phase flow Sample size (FIA and SEC)	0.4 mL min^−1^ 50 µL	Fragmentor voltage	120 V
Monitored isotopes	^181^Ta, ^197^Au	Scanning range, m/z	50–1000
Internal standard	^103^Rh	Scanning step	0.1 amu
Integration time	0.1 s	Dwell time	0.2 s
**Separation conditions**			
**Reversed-phase liquid chromatography (RPLC)**		**Capillary zone electrophoresis (CZE)**	
Parameter	1200 Capillary HPLC	Parameter	G1600AX 3D CE System
Column type	PFP (Phenomenex), 150 × 1 mm, 3 µm	Capillary type, length, and ID	Fused silica, 100 cm, 75 µm
Flow	5 µL·min^−1^	Voltage	20 kV
Sample volume	1–8 µL	Sample volume	50 mbar × 12 s (100 nL)
Mobile phase composition	A: 10 mM ammonium acetate, pH = 7.0 B: MeOH	Background electrolyte, BGE	20 mM ammonium acetate, pH = 6.9
Elution method	Gradient: 5 min—0% B 20 min—90% B 25 min—90% B	Capillary conditioning and washing according to previous protocols [6] Each step took 5 min	Flash with 1M NaOH Flash with H_2_O Wait Flash with H_2_O Flash with BGE

## Data Availability

Data is contained within the article and Appendix A.

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
