# Peer review of "Speciation Analysis Highlights the Interactions of Auranofin with the Cytoskeleton Proteins of Lung Cancer Cells"

_pharmaceuticals, 2022, doi:10.3390/ph15101285_

Round 1

Reviewer 1 Report

The manuscript 'Speciation analysis highlights the interactions of auranofin with the cytoskeleton proteins of lung cancer cells' by M. Kupiec et al. presents a very thorough study of the interaction of a very important anticancer drug -auranofin- with two cell lines, A-549 and MRC-5.

The authors designed very well the research by starting with separating hydrophilic and hydrophobic compounds, followed by SEC analysis, then trypsin digestion for identifying binding peptides (that will lead them to the parent proteins) concluding that most candidate proteins are cytoskeleton proteins. Last, they performed a wound healing test with auranofin and comparison with cisplatin.

Overall, I am in favor of publication after commenting the following:

-the are some statements regarding cisplatin (lines 142 and 145) however neither the data nor the actual values are given. These statements should either be amended or the data should be shown (in SI).

-define AC and DC in fig. 2

-define PCA when describing the chart

Author Response

Detailed responses to the reviewers of the manuscript pharmaceuticals-1966896 “Speciation analysis highlights the interactions of auranofin with the cytoskeleton proteins of lung cancer cells” by Katarzyna Pawlak

Reviewer 1

STYLE: Reviewer Comment - Response

Gen_Com: The manuscript 'Speciation analysis highlights the interactions of auranofin with the cytoskeleton proteins of lung cancer cells' by M. Kupiec et al. presents a very thorough study of the interaction of a very important anticancer drug -auranofin- with two cell lines, A-549 and MRC-5.

The authors designed very well the research by starting with separating hydrophilic and hydrophobic compounds, followed by SEC analysis, then trypsin digestion for identifying binding peptides (that will lead them to the parent proteins) concluding that most candidate proteins are cytoskeleton proteins. Last, they performed a wound healing test with auranofin and comparison with cisplatin.

We are very thankful for the revision of the manuscript and helpful comments. Please read the detailed responses below. We have also attached the file with the changes indicated in the manuscript.

Overall, I am in favor of publication after commenting the following:

Comment 1:-the are some statements regarding cisplatin (lines 142 and 145) however neither the data nor the actual values are given. These statements should either be amended or the data should be shown (in SI).

RE 1. We have added Figure S1 made analogous to Figure 2 for the gold distribution. The figure includes the relative amounts for platinum distribution to compare with gold.

Comment 2:-define AC and DC in fig. 2

RE 2. The “AC” and “DC” was added in the title of figure 2.

Comment 3:-define PCA when describing the chart

RE 3. “Principal component analysis” was added in the text and title for figure 6.

Reviewer 2 Report

Some gold complexes were reported with remarkable antiproliferative potency. Auranofin is a well-known therapy prodrug for rheumatoid arthritis. The activity and mechanism of action for auranofin toward cancer cells is still poorly understood. It is necessary to analyze the possible action mechanism of Auranofin from the view point of the metallomics and proteome. The studies and results in this manuscript will be interested to the researchers in the fields of bioinorganic chemistry and medical chemistry. However, some revisions should be considered.   

1.  “Auranofin shows a better affinity to proteins than DNA/RNA in contrast to cisplatin.” Cisplatin is classical anticancer drug, it is one of the best model complex for comparison. More references about cisplatin could be cited and detail comparison with cisplatin could be discussed.

2.  In Title, address:“Chair” of Analytical Chemistry, Faculty of Chemistry, should be checked. Is it "Department" ?

3.  In Figure 5 and Figures 6b and 6e, the font of labels in X axis and Y axis is too small, it should be enlarged.

4.  The language should be refined.

Author Response

Reviewer 2

STYLE: Reviewer Comment - Response

Gen_com: Some gold complexes were reported with remarkable antiproliferative potency. Auranofin is a well-known therapy prodrug for rheumatoid arthritis. The activity and mechanism of action for auranofin toward cancer cells is still poorly understood. It is necessary to analyze the possible action mechanism of Auranofin from the view point of the metallomics and proteome. The studies and results in this manuscript will be interested to the researchers in the fields of bioinorganic chemistry and medical chemistry. However, some revisions should be considered.   

We are very thankful for the revision of the manuscript and helpful comments. Please read the detailed responses below. We have also attached the file with the changes indicated in the manuscript.

Comment 1: “Auranofin shows a better affinity to proteins than DNA/RNA in contrast to cisplatin.” Cisplatin is classical anticancer drug, it is one of the best model complex for comparison. More references about cisplatin could be cited and detail comparison with cisplatin could be discussed.

Ad. 1. According to the Reviewer's request, we have added results obtained for cells exposed to cisplatin. It is presented in figure S1. Moreover, we added five References [41-5] discussing cisplatin activity and an additional passage related to comparing both drugs concerning Figure 2-3, and the Conclusion section.

Page 4, added passage: “In normal cells, platinum compounds were present in non-adherent cells and were evenly distributed (35%, 37% and 27% of platinum was determined in the fraction containing hydrophobic, protein and hydrophilic compounds, respectively). This is in agreement with the results of other research teams showing that cisplatin, in addition to its affinity for DNA [41], forms stable adducts with cysteine- and methionine-containing proteins [42-43], shows an affinity for copper transport proteins and affects the structure of the hydrophobic plasma membrane (PM) [44]. Both metal complexes cisplatin and auranofin show affinity for a similar group of compounds in A-549 cells. In the case of fibroblasts (MRC-5), the gold content was detected proportionally to the number of adherent and non-adherent cells (92% of the total cell population consists of adherent cells that contain 98% of total accumulated gold). Unlike platinum content concentrated in non-adherent cells (50% of the total cell population consists of non-adherent cells that contain 99% of total accumulated platinum). This indicates the homogeneous accumulation of AUF’ by the fibroblast cells and the drug-induced uptake of cisplatin. Cisplatin was proven to interact with proteins and lipids, influencing membrane composition, fluidity, and permeability [44-45]. Such alteration of PM can enhance metallodrug diffusion and uptake. It depends on the cell type due to differences in PM and ECM composition [45] (Fig. S1).”

Other changes are highlighted in the manuscript.

Comment 2: In Title, address:“Chair” of Analytical Chemistry, Faculty of Chemistry, should be checked. Is it "Department" ?

RE 2: We understand the confusion. Nevertheless, it is correct. It is Chair, not the Department. Please check the faculty website for the Structure of our Institution: https://www.ch.pw.edu.pl/ch_en/About-us/Structure

Comment 3: In Figure 5 and Figures 6b and 6e, the font of labels in X axis and Y axis is too small, it should be enlarged.

RE 3. We apologize. The font size is indeed inconvenient to read. We corrected the font size in both figures

Comment 4:  The language should be refined.

RE 4. Appropriate corrections of English style are indicated in the manuscript (Word->registration of corrections).